# Hydroxypropyl Methylcellulose Capsules Enhance Aerodynamic Performance of Carrier-Based Dry Powder Inhaler Formulations: A Comprehensive Evaluation of Capsule Material Effects [note 1]

**DOI:** 10.3390/pharmaceutics17121621

**Published:** 2025-12-17

**Authors:** Camille Dumont, Sandrine Picco, Beatriz Noriega-Fernandes, Pierre Verlhac, Andrea Elena Cortez, Camille Boulet, Molly Gallagher, Christopher Bock, Vincent Jannin

**Affiliations:** 1Capsugel France SAS—Lonza, 68000 Colmar, France; camille.dumont@lonza.com (C.D.);; 2Lonza Advanced Synthesis, Lonza Group AG, Bend, OR 97701, USA; beatriz.fernandes@lonza.com (B.N.-F.);

**Keywords:** dry powder inhalation, capsule, aerodynamic particle size distribution, carrier-based formulation

## Abstract

**Background/Objectives**: This study aims to investigate the underexplored impact of capsule type on the performances of capsule-based dry powder inhalers (cDPIs). It compares specific properties of hard gelatin-based capsules (Hard Gelatin Capsules (HGC), HGC including polyethylene glycol (HGC + PEG)) and hypromellose-based capsules, (Zephyr^®^ Vcaps^®^ (VC), Zephyr^®^ Vcaps^®^ Plus (VCP) and Vcaps^®^ Plus Zephyr Inhance™ (VCP-I)) with aerosolization performances of model carrier-based formulation, providing insights into their impact on pulmonary drug delivery efficacy. **Methods**: Aerosolization properties of a model phenytoin/lactose blend formulation filled in the different capsules was evaluated using a Next Generation Impactor (NGI) with RS01 device. Capsule shell characteristics were evaluated in terms of water activity, static charges, and inner surface aspect and roughness. **Results**: Hypromellose-based capsules, especially VC and VCP-I, exhibited significantly higher drug delivery performances compared to gelatin-based capsules. In particular, VCP-I demonstrated good results with excellent batch-to-batch reproducibility and 51% of the nominal dose available for lung absorption. Although capsule inner surface showed clear differences between both polymer families, no clear correlation could be found between cDPI performances and capsule roughness and density of charge. All capsules presented good mechanical properties in the conditions of the tests. **Conclusions**: Capsule type exerts a significant impact on cDPI performances. These findings highlight the importance of capsule selection as a critical material attribute in the design and optimization of inhalation products.

## 1. Introduction

Inhalation products are widely used to treat and prevent lung infections such as asthma and chronic obstructive pulmonary disease. Among the different types of existing delivery systems for pulmonary administration, dry powder inhalers (DPIs) are extensively used owing to their accuracy, stability, and facility of use. In particular, capsule-based DPIs (cDPIs) present the advantage of dose flexibility and low development manufacturing costs with a wide offer of off-the-shelf devices.

The performance of cDPIs is influenced by many factors. Among these, the importance of formulation is well documented, with numerous studies addressing the impact of the type of formulation (carrier-based or carrier-free formulation) excipient selection or importance of API micronization [1,2,3,4]. The impact of devices on the efficacy of cDPIs is also reported in several scientific articles [5,6,7,8]. However, although the interdependence between formulation, device, and capsule on pulmonary drug delivery is often mentioned, the impact of the capsule itself is relatively under-investigated. Yet formulators can choose from a variety of capsule types, and this choice can substantially influence cPDI performance.

Hard gelatin capsules (HGCs), traditionally produced by dipping cold metal pins into hot gelatin solutions, remain widely used in cDPIs. However, their relatively high water content (13–16%) makes them prone to brittleness and cracking under the low relative humidity conditions typical of DPI environments. To overcome this potential limitation, plasticizers such as polyethylene glycol (PEG) have been added to gelatin to improve mechanical flexibility of the capsules. Later, hypromellose (HPMC) capsules, containing reduced water content (3–9%) have emerged as promising alternatives. These HPMC capsules can be manufactured by the traditional cold-gelation method, including a gelling agent in the polymer formulation preparation, as for Zephyr^®^ Vcaps^®^ capsule (VC) and Quali-V^®^ I, or fabricated using a thermo-gelation process without gelling agent. This method consists in dipping hot metal pins in a cold HPMC solution and is used to manufacture Zephyr^®^ Vcaps^®^ Plus (VCP) and Vcaps^®^ Plus Zephyr Inhance™ capsule (VCP-I). The latter capsule is obtained with an improved manufacturing process enabling reduced capsule variability.

Low water content is particularly advantageous for DPIs because it helps preserve formulation stability and ensures consistent powder release [9]. Environmental conditions, particularly temperature and relative humidity (RH), significantly influence the moisture content and mechanical behavior of capsule polymers, which in turn affects their physicochemical properties and critical aerosol performance metrics. The polymer chemistry and water sorption characteristics of HPMC and gelatin capsules create distinct responses to environmental stress, with implications for aerodynamic particle size distribution (APSD), fine particle fraction (FPF), and emitted dose consistency [10]. While the different HPMC capsule variants are recognized for their beneficial characteristics in terms of low water content and improved control over capsule mechanical properties (avoiding brittleness) [11], their impact on DPI aerosolization performance and drug lung delivery remains insufficiently understood.

Recent investigations highlight that capsule material (e.g., gelatin versus HPMC) significantly influences key performance attributes: for instance, HPMC capsules typically exhibit lower moisture content and greater mechanical robustness under low-humidity conditions, translating into improved capsule puncture behavior, reduced powder retention and enhanced fine-particle dose when compared to gelatin shells [12]. Moreover, capsule tribo-electrification behavior, aperture geometry (size, number, orientation of puncture pins), and capsule-chamber motion dynamics have been shown to interplay with powder properties and device flow regimes—leading to differences in aerosolization efficiency and dose reproducibility [13]. Hence, the capsule is not merely a passive container but a critical component influencing metering, release, dispersion, and ultimately lung deposition—a fact that warrants systematically including capsule properties in the quality-by-design framework of cDPI development [6].

This study investigates the underexplored influence of capsule type on the aerodynamic performance of a model carrier-based DPI formulation used with a well-established inhalation device. A complete capsule portfolio was evaluated in parallel using complementary techniques to assess both performance and mechanical properties and to correlate capsule-specific characteristics with drug delivery behavior. The novelty of this work lies in the first systematic evaluation of the newly developed VCP-I capsule, manufactured using an improved thermo-gelation process, and in assessing its performance relative to VCP and cold-gelled capsules. This integrated approach provides new insight into the critical role of capsules in DPI performance.

## 2. Materials and Methods

### 2.1. Materials

Phenytoin was purchased from Spectrum Chemical Mfg. Corp. (New Brunswick, NJ, USA). Respitose^®^ SV003 (coarse lactose) and Lactohale^®^L230 (fine lactose) were obtained from DFE Pharma (Goch, Germany). Size 3 DPI grade Capsugel^®^ Zephyr^®^ capsules, Hard Gelatin Capsules (HGC), Hard Gelatin Capsules with 5% PEG (HGC + PEG), Vcaps^®^ capsules, Vcaps^®^ Plus capsules, and 3 batches of Vcaps^®^ Plus Inhance™ capsules were used, and their characteristics are listed in Table 1. RS01 high-resistance devices were kindly provided by Plastiape (Osnago, Italy).

Ultra purified water was produced on-site by milliQ-systems. Methanol and acetonitrile were acquired by Biosolve (Dieuze, France). Isopropanol, ammonium phosphate dibasic, phosphoric acid 85%, ammonium hydroxide, and hexane were purchased from Merck (Darmstadt, Germany). Silicone oil was provided by Dow (Midland, MI, USA).

### 2.2. Methods

#### 2.2.1. Formulation Preparation

Phenytoin was micronized using an MC50 spiral jet mill under controlled conditions: feeding pressure of 5.5 bar, grinding pressure of 4.5 bar, and a feeding rate of 0.1 kg/h. This process aimed to achieve a target particle size distribution within the inhalation range (1–5 µm).

A model formulation was prepared by blending 125 g of a ternary mixture. The composition of this mixture was 79.5% *w*/*w* Respitose SV003 (coarse lactose), 15% *w*/*w* Lactohale LH230 (fine lactose), and 5.5% *w*/*w* micronized phenytoin. Prior to blending, both lactose grades and the API were sieved through a coarse µm sieve.

The two lactose grades were subsequently pre-blended for 30 min at 32 RPM in a 1-quart Patterson-Kelley blender (Buflovak, Tonawanda, NY, USA). To ensure a homogeneous powder blend, the API was then incorporated into the pre-blend using a two-step process with the same equipment: (1) half of the API was blended with half of the pre-blend for 30 min at 32 RPM; (2) the remaining API and pre-blend were then processed for an additional 60 min under the same conditions.

#### 2.2.2. Capsule Filling

To equilibrate their moisture content prior testing, all capsules were stored for at least 10 days in desiccators at 45% RH levels at room temperature (20 °C ± 2 °C). Saturated solution of potassium carbonate was used to achieve 45% RH.

Capsules were then manually filled with 25.0 mg of formulation, corresponding to 1.38 mg of phenytoin. Each capsule type was filled less than 7 days before evaluating the aerodynamic particle size distribution, with a minimum of 2 days equilibration at 45% RH before initiating the test.

#### 2.2.3. Formulation Characterization

##### Particle Size Distribution

Wet particle size distribution of micronized phenytoin was determined by laser diffraction using a Malvern Mastersizer 3000 (Westborough, MA, USA) equipped with Hydro MV wet dispersion unit. Approximately 30 mg of sample was dispersed in 20 mL of 0.1% *w/v* solution of Span^®^ 85. The dispersion was continuously circulated at 2500 RPM and sonicated during analysis to ensure complete deagglomeration. Measurements were performed when obscuration was stable between 7% and 16%. Each sample was analyzed in triplicate (*n* = 3).

##### Solid-State Characterization

The solid-state properties of the micronized material were characterized to confirm its crystalline state.

X-ray Powder Diffraction (XRPD) patterns of the micronized API were collected on a Bruker D2 Phaser diffractometer (Billerica, MA, USA) (Cu Kα radiation; 30 kV, 10 mA). Samples were prepared by leveling powder into a standard sample holder. Scans were performed from 3° to 50° (2θ) with a step size of 0.02° (2θ).

Dynamic Vapor Sorption (DVS) analysis was performed on the micronized material using a Q5000 SA (TA Instruments, New Castle, DE, USA) to assess moisture sorption/desorption and potential amorphous content. Data was evaluated with Universal Analysis 2000 software.

##### Mixture Morphology

Scanning Electron Microscopy (SEM) analysis of the powder blend morphology was performed using a SU3500 (Hitachi, Hillsboro, OR, USA) microscope. Samples were prepared on an adhesive stub and coated under vacuum with gold/palladium alloy. Imaging was conducted at 15 kV (working distance: 1 mm) at magnifications 500× to 1500×.

##### Water Activity

Water activity of the formulation and capsules were measured with a Rotronic HC2A-AW probe (Process Sensing Technologies PST SAS, Saint Priest, France) with PS-40 and WP-40 sampler. About 1 g of formulation and about 3 g of capsules were used for each test. Each measurement was performed in triplicate (*n* = 3).

#### 2.2.4. Aerodynamic Particle Size Distribution

The aerodynamic particle size distribution (APSD) of the model formulation released from tested capsules and the RS01 device was evaluated with a Next Generation Impactor (NGI, Copley Scientific, Nottingham, UK) equipped with a pre-separator. All pans were coated with a solution of 1% silicon oil in hexane prior testing.

For each test, one filled capsule was pierced in the RS01 device and its content was vacuumed by a 60 L/min flow rate (4 kPa pressure drop). Phenytoin was collected in the capsule, the device, and in each section of the NGI (pre-separator, mouthpiece, cups 1 to 7 and micro-orifice collector) using a solution composed of 50 mM ammonium phosphate buffer pH 2.5/methanol/acetonitrile (65:21:14) and quantified with an appropriate HPLC-UV method (see Section 2.2.5). Each test was performed in triplicate, on three consecutive days, in a laboratory with controlled RH and temperature (RH of 50% ± 5%, RT of 22 °C ± 2 °C).

Recovery was defined as the fraction of the nominal phenytoin quantified in all the sections of the test (including capsule and device).

Emitted Dose (ED, mg) was defined as the amount of phenytoin released from the device and the capsule. Emitted Fraction (EF, %) was expressed as the ED divided by the recovery.

The Fine Particle Dose (FPD, mg) was defined as the amount of phenytoin collected in the NGI with particle size below 5 µm. This result was divided by the ED to express the Fine Particle Fraction (FPF,%).

#### 2.2.5. Phenytoin Quantification Method

Phenytoin was quantified by High Performance Liquid Chromatography (HPLC, Waters, Guyancourt, France) systems equipped with a UV-DAD detector (Waters). Separation was made on Symmetry C18, 3.5 µm, 2.1 mm × 50 mm column at 40 ± 5 °C. Samples were stored in an autosampler regulated at 5 ± 5 °C. An isocratic elution was applied, using a mobile phase composed of 50 mM ammonium phosphate buffer pH 2.5/Methanol/Acetonitrile (65:21:14). The flow rate was 0.4 mL/min, the injection volume was 10 μL, and the wavelength of the detector was set at 220 nm. The acquisition duration was 8 min, and phenytoin eluted at 3 min. Limit of quantification was 0.05 µg/mL.

#### 2.2.6. Capsule Observations

Capsule characterization tests were performed on capsules equilibrated at the same conditions as described in Section 2.2.2.

##### Macroscope

Filled capsules were pierced using a RS01^®^ device (Plastiape), and emptied in a Dosage Unit Sampling Apparatus (DUSA, Copley Scientific) using the same parameters used for the NGI test (i.e., 60 L/min for 4 s). The pierced holes on both body and cap were observed with a Leica MZ12 5 microscope at a ×10 magnification.

##### Scanning Electronic Microscopy

Scanning Electron Microscopy (SEM) was performed with a FlexSEM1000-SU1000 microscope (Hitachi). Filled capsules were emptied in a DUSA device using a RS01^®^ as the device and the same parameters used for the NGI test (i.e., 60 L/min for 4 s). Emptied capsules were then cut across the whole body length using scissors to explore the internal surface of emptied capsules. Samples were deposited on a flat steel holder on a graphite sticker and coated under vacuum by cathodic sputtering with a layer of 5 nm of gold/palladium alloy. The samples were observed by SEM technique under an acceleration voltage of between 5 and 10 kV. The same conditions were used to observe the inner surface of empty capsules.

##### Capsule Inner Roughness

The roughness of the inner surface of the capsules was assessed parallelly (axial) or perpendicularly (radial) to the length of the capsules with a MarSurf SD26 (Mahr, Göttingen, Germany). The BFW A 10-45-2/60 probe was used in the axial direction. In that case, the exploration length was 5.6 mm. The measurements were repeated 3 times on 3 different capsules. In the radial direction, the BFW A 0.7-45-2/90 probe was used and the exploration length was 0.56 mm to probe a flat surface. To compensate for the small exploration length and probe more surface, the number of repeats for each measurement was increased to 6 on 3 different capsules. Results are expressed as Rz (µm), which is the average peak to valley distance on five sequential sampling lengths within the measuring length. The capsules analyzed were randomly picked within batches described in Table 1.

##### Static Charges

To evaluate the charging tendencies of the capsules stored at 45% RH, a GranuCharge™ (Granutools, Awans, Belgium) equipment was used in a controlled room (RH of 50% ± 5%, RT of 22 °C ± 2 °C). Measurements were performed as follows: The initial charge density (q0) of between 5 and 10 g of capsules was evaluated by pouring them into the Faraday cup of the equipment after they had been linked to the mass, i.e., uncharged for 1 min. The capsules were then left to rest for 1 min during which they were linked to the mass Then, they were poured during 15 s (feeding time) through a V-tube vibrating at 50%. The capsules dropped into the Faraday cup where the density of charge was measured again (q1). ABS plastic V-tube (same material as RS01^®^ device) was used in this measurement, with an outer diameter of 5 cm disposed with an inclination of 45° and arranged at a 90° angle with an overall length of 70 cm. The charges obtained were normalized by the mass of capsules. The relative density of charge was obtained directly in the software by subtracting the initial charge from the final charge (after contact with the V-tube q1-q0). Tubes were cleaned with a dry brush and washed with ethanol and let dry completely between each analysis. Measurements were performed in triplicate for each capsule type.

#### 2.2.7. Statistical Analysis

Statistical analyses on APSD results were carried out by first checking the homoscedasticity of variance using Levene’s test. The normality of the data residues was assessed graphically. Likewise, the normality of the data distribution obtained for the rugosity and the electrostatic charge of the samples was assessed using the Shapiro–Wilk test.

For APSD, statistical comparisons on the mean of responses between independent groups of data were performed using a one-way analysis of variance (ANOVA, *n* = 3). When the ANOVA multi-group test was significative, post hoc tests (HSD Tukey) were used to highlight group pairs of interest. When the homoscedasticity of the variance was not confirmed, then the Welch’s test and the Games–Howell tests were considered to assess whether the capsule had an effect and, if so, to determine whether these effects were significantly different.

## 3. Results

### 3.1. Formulation Characterization

Micronized phenytoin exhibited a particle size distribution characterized by a Dv50 of 1.8 µm and a Dv90 of 3.3 µm (RSD ≤ 10%), consistent with an inhalation range.

XRPD and DVS analyses were performed to evaluate the impact of the micronization process on the solid-state properties of the material. (Figure 1). Furthermore, DVS analysis did not detect a weight change when exposing the material above the critical relative humidity of 50%, as presented in Figure 2, indicating amorphous phenytoin was not detected in the micronized sample. Collectively, these solid-state characterization results suggest that the micronized material is stable and suitable for subsequent formulation activities aimed at developing a stable product.

Following blending, mixtures were characterized by Scanning Electron Microscopy (SEM) to assess particle morphology and interaction (Figure 3). SEM images revealed that the fine particles formed agglomerates and also exhibited interaction with the coarse lactose surface.

### 3.2. Aerosolization Properties

#### 3.2.1. Impact of Capsule Type on Aerosolization Properties

Phenytoin recovery after evaluation of the APSD with NGI was between 80% and 120% (85% to 117%) of the nominal dose. Therefore, API distribution profiles in the NGI, tested with various capsule types, were expressed as percentage of the nominal dose, Figure 4. Although the same trends can be observed, capsule types seem to influence phenytoin distribution in the different sections of the apparatus. In particular, API quantification in the capsule shows higher retention in gelatin-based capsules.

On the other hand, phenytoin quantified in capsules was significantly lower for HPMC-based capsules and, in particular, for VC and VCP-I, Figure 5a. This phenomenon has a direct impact on EF, with significantly higher values calculated for VC and VCP-I, while VCP behaved similarly as gelatin-based capsules, Figure 5b.

FPF was not influenced by capsule type, Figure 6a. However, when the impact of capsule retention is taken into account by dividing the FPD by the nominal dose, both VC and VCP-I capsules show higher levels of phenytoin available for lung absorption, statistically different from HGC capsules. Only VCP-I was statistically different from both types of gelatin-based capsules, with higher value compared to VC (51 ± 6% vs. 47 ± 7%) which confirms superior performance of this capsule type for the tested formulation. For VC and VCP-I, a higher variability in FPD/nominal was observed compared to the three other reference capsules. This increased variability can be attributed to the higher variability in API recovery during NGI testing for these two capsule types. As recovery is not directly correlated with the capsule material, and given the low variability observed in phenytoin retention within the capsules (Figure 5a), this behavior is not considered to be capsule-dependent.

#### 3.2.2. Evaluation of VCP-I Robustness

Impact of inter-batch variability on DPI performance was evaluated with VCP-I which was recently marketed and whose performance has not yet been published. In addition, this capsule presented the highest FPD/nominal value. To this aim, APSD measurements were performed by NGI on two additional batches: VCP-I B and VCP-I C. NGI distribution profiles, presented in Figure 7, show a similar profile for the three batches. Slightly higher values were obtained with VCP-I which can be explained by a highest recovery (97–117%) compared to VCP-I B (85–103%) and VCP-I C (89–101%).

No statistical differences were observed regarding phenytoin retention in capsule, EF, FPF, and FDP/nominal, Table 2, confirming VCP-I robustness.

### 3.3. Capsule Characterizations

#### 3.3.1. Capsule Water Activity

Water activity of the formulation was measured at 0.48 ± 0.01. Therefore, capsules were equilibrated at 45% RH to limit water exchanges between the capsules and the fill formulation. After 10 days in these RH conditions, capsules presented a water activity around 0.43, confirming that they were well equilibrated, Table 3.

#### 3.3.2. Macroscope Observations

Observations of capsules emptied from the model formulation show powder retention at the surface of capsule shells, especially for transparent capsules. However, these observations are not necessarily correlated with phenytoin quantified in the capsules after NGI test, as it seems that similar retention is observed in VCP-I and VC while a difference was measured by HPLC after NGI test, Figure 8.

Observations of the puncturing holes show regular spherical holes for all capsule types, with flaps remaining attached to the capsule parts (body and cap).

#### 3.3.3. Inner Capsule Surface Observations by SEM

Observations of empty capsules shells show important differences between the two types of polymers. Indeed, droplets can be seen at the surface of gelatin-based capsules giving the impression of a very heterogenous surface, Figure 9. On the other hand, HPMC-based shells seem smoother although oriented stripes are more pronounced, particularly for both VCP and VCP-I capsules.

These observations were performed on a limited number of capsules, all produced on different capsule manufacturing lines and on different metal pins with their own level of wear. Therefore, these observations may not be representative of the entire batches.

SEM observations of the capsule inner surface after emptying from the carrier-based formulation also show differences between HPMC-based capsules and gelatin-based with higher particle retention on the latter. Regarding HPMC-based capsules, less powder appears to be retained on VCP and VCP I compared to VC.

#### 3.3.4. Capsule Roughness

Roughness results were impacted by the direction of the measurement, with higher values obtained in the axial direction (between 1.02 ± 0.06 µm for HGC and 0.48 ± 0.06 µm for VCP) compared to the radial direction (between 0.27 ± 0.09 µm for HGC + PEG and 0.51 ± 0.16 µm for VCP I), Figure 10. This can be explained by the slight curvature of the sample in axial direction, for which the exploration length had to be shortened to avoid the curvature of the capsules. Nevertheless, these values remained low, representing less than 1% of the average capsule thickness (about 100 µm).

There was not any clear impact of the polymer type or the HPMC capsule manufacturing process (gelling agent vs. thermo-gelation) on the surface roughness.

#### 3.3.5. Capsule Electric Charges

The density of charges of gelatin-based capsules equilibrated at 45% RH, and was significantly lower than the density of charges of VC and VCP capsules, equilibrated at similar RH, Figure 11. The density of charges of VCP-I, 0.66 ± 0.56 nC/g, was in line with gelatin-based capsules with 0.81 ± 0.13 nC/g and 0.72 ± 0.09 for HGC and HGC + PEG, respectively. However, results obtained with VCP-I presented more variability than the other capsule types.

## 4. Discussion

This study demonstrates that capsule type significantly influences the aerodynamic performance of carrier-based dry powder inhaler formulations, with HPMC-based VCP-I capsules exhibiting superior drug delivery efficiency compared to gelatin-based capsules. VCP-I capsules achieved the lowest API retention (6.30 ± 0.23% of nominal dose) with minimal variability, resulting in 51% of the encapsulated phenytoin being available for lung absorption when used with the RS01 device.

While capsule-based DPI performance depends on numerous interconnected factors, including formulation composition (API particle size and physicochemical state, and carrier selection and particle size distribution), manufacturing processes (micronization conditions, blending parameters, and capsule filling), device characteristics (piercing mechanisms, airflow resistance, and chamber geometry), and patient variables (inhalation flow rate, and inspiratory capacity), the capsule component itself has received disproportionately limited scientific attention despite its potential impact on drug delivery efficiency. This knowledge gap provided the rationale for systematically evaluating capsule type effects on aerodynamic behavior using a well-characterized carrier-based formulation and the standardized RS01^®^ device.

Carrier-based DPI formulations rely on precisely controlled cohesive and adhesive interactions between micronized API particles (1–5 μm), coarse lactose carrier particles (50–150 μm), and fine excipient particles that modulate inter-particulate forces. Within this framework, capsule material properties introduce additional interaction mechanisms that significantly influence overall performance through powder adhesion to interior surfaces during aerosolization. The capsule interior presents substantial contact area where particles can adhere via the same fundamental force mechanisms governing drug–carrier interactions, with material-specific differences in polymer chemistry, surface energy, moisture content, and manufacturing-induced surface characteristics affecting powder retention and release consistency.

### 4.1. Primary Performance Findings

Capsule type influence on the aerodynamic performances of the model formulation was evaluated by NGI. Results show that API retention in the capsule was influenced by the type of capsule. The performance hierarchy observed (VC ≥ VCP-I > VCP > gelatin-based capsules) for API retention represents a clinically meaningful difference in drug delivery efficiency. Importantly, this study provides the first evaluation of the newly developed VCP-I capsule, which has not previously been tested. The comparable performance of VCP-I to VC demonstrates that thermo-gelled capsules manufactured using an improved fabrication process can achieve performance equivalent to cold-gelled VC capsules. These results are consistent with those obtained in a study using a ternary blend formulation with 0.05% formoterol, stored at 50% RH, where the same capsule ranking was observed, although the impact of the capsule evaluated was on 10 capsules per test (*n* = 3) [13]. With gelatin-based capsules retaining 12–15% of the nominal dose compared to 3–6% for optimal HPMC capsules, this translates to approximately 10% more active pharmaceutical ingredient being available for pulmonary delivery. For respiratory medications where precise dosing is critical for therapeutic efficacy and safety margins, this improvement could significantly impact clinical outcomes.

Capsule retention has a direct impact on EF, corresponding to the fraction of dose released from the capsule and device, for which higher values were obtained with VC followed by VCP-I, VCP, and then both gelatin-based capsules.

Concerning the FPF, no influence of the capsules was observed. This result was expected as FPF corresponds to the fraction of phenytoin from the ED with particle size below 5 µm, and therefore only takes into account what is released from the cDPI. The similarity between FPF values obtained for all capsule types highlights the robustness of the formulation, which aerosolized similarly in the NGI apparatus with a final FPF between 51% and 57%.

On the other hand, when the FPD is related to the nominal dose, capsule type has a clear influence on the results with higher values obtained for VC and VCP-I. Because of the lower variability observed with VCP-I, it can be considered as outperforming the other capsule types, enabling 51% of the encapsulated phenytoin to be available for lung absorption. VCP-I would then be recommended for the development of a cDPI product of this model phenytoin carrier-based formulation using a RS01^®^ device.

Capsule overall appearance after powder emptying show powder retention in all capsule types and is obviously more visible in transparent capsules. With these particular capsules, it seems that powder retention is higher in VCP-I followed by VC and then HGC which is not correlated with phenytoin quantification in capsules after NGI evaluation (VC > VCP-I > HGC). This reflects the importance of API quantification to evaluate the performance of cDPI.

### 4.2. Mechanistic Understanding of Capsule Performance Differences

The superior performance of HPMC capsules, particularly VCP-I, stems from fundamental differences in polymer chemistry, surface characteristics, and manufacturing processes that influence powder–capsule interactions through multiple mechanisms

Observations of capsule inner surface after emptying are in accordance with NGI results with more powder retained on the surface of gelatin-based capsules compared to HPMC-based capsules.

SEM revealed distinct morphological differences between capsule types. Gelatin-based capsules exhibited heterogeneous surfaces with droplet-like structures. In contrast, HPMC capsules presented smoother surfaces with a more uniform appearance. However, these observations could not be correlated with rugosity measurements, which show high variability and do not clearly discriminate the different capsule polymers and fabrication processes. No parallel could be established with NGI results.

A study performed by Saleem et al. used atomic force microscopy to map the inner surface of HPMC capsules with various lubricant content. This study reports that lubricant content can smoothen inner surface of capsules and facilitate API emission upon inhalation [14]. In the present study with phenytoin, and contrary to conventional understanding, lubricant content did not correlate directly with capsule performance. Although different lubricants are used across the capsule types evaluated (depending on the polymer and fabrication process) this diversity prevents any direct comparison of lubricant composition or its impact on performance between all capsule types. The only meaningful comparison can therefore be made between VCP and VCP-I, which share the same polymer and fabrication process. In VCP-I, the thermo-gelation fabrication process has been optimized to reduce the amount of deposited lubricant (VCP-I = 46 ppm vs. VCP = 174 ppm). Despite having the lowest lubricant content, VCP-I demonstrated excellent performance and is even comparable to that of VC capsules fabricated with a cold-gelation process (209 ppm) generally recognized as performing better than capsules fabricated by thermo-gelation process [13]. This suggests that lubricant composition, distribution, and integration within the polymer matrix, rather than absolute content, may be more critical for performance. The improved manufacturing process used for VCP-I capsules results in more uniform lubricant distribution, explaining the superior capsule performance despite lower total lubricant content.

Environmental humidity conditions critically influence both capsule physicochemical properties and aerodynamic performance metrics. Studies have demonstrated that exposure to elevated humidity (75% RH at 40 °C) can result in up to 50% reduction in FPD for certain DPI products, with corresponding decreases in FPF due to moisture-induced particle agglomeration and altered inter-particulate forces [15]. On a capsule perspective, recent findings by Magramane et al. (2025) confirmed that under elevated humidity (25 °C, 75% RH), HPMC capsules maintain their mechanical integrity and puncture resistance, while gelatin capsules soften, plasticize, and suffer a marked decline in structural stability [12]. Their microstructural analysis (via positron annihilation lifetime spectroscopy) revealed that HPMC capsules undergo dynamic free-volume expansion, in contrast to the more rigid semi-crystalline structure of gelatin [12]. In addition, a seminal study by Benke et al. (2021) directly compared gelatin, gelatin-PEG, and HPMC capsules in the context of ciprofloxacin hydrochloride DPI formulations [16]. They carried out a long-term stability study (6 months, ICH conditions) and found that HPMC capsules provided the greatest formulation stability and the best in vitro aerodynamic performance, relative to both gelatin and gelatin-PEG shells. Specifically, they observed that the residual solvent content of the HPMC capsules decreased the least over time, helping preserve capsule integrity, reduce fragmentation, and ultimately improve lung deposition—with a ~10% higher deposition in the in vitro lung model compared to the other capsule types [16]. Nevertheless, it is common for DPI applications to work under much lower RH because of API and formulation sensitivity to water (i.e., spray-dried formulations). In this case, capsule flexibility may impact the emitted dose due to an increased risk of capsule brittleness. In particular, storage below 40% RH increases gelatin capsule brittleness, potentially compromising puncture performance and dose reproducibility. This capsule moisture sensitivity directly impacts critical quality attributes including aerodynamic particle size distribution; hygroscopic excipients exposed to elevated humidity form stronger agglomerates through increased capillary forces, resulting in reduced fine particle fraction and altered lung deposition profiles.

Capsules based on gelatin and HPMC do not possess the same moisture content with, respectively, 13 to 16% and 3 to 9% [11]. In the present study, all capsules were all carefully equilibrated at 45% RH consistent with the water activity of the formulation. Therefore, potential water transfer from the capsule to the formulation cannot justify the differences in capsule retention observed between the different capsule types. Moreover, because at 45% RH all capsule types were in their recommended conditions of use (from 35 to 65% RH between 15 and 25 °C), none of them exhibited brittleness upon puncturing and all presented regular puncturing holes with flaps remaining well attached to the capsule parts. Capsule brittleness was therefore not responsible for the differences observed in NGI. The selected RH condition falls in the classical range of previously published conditions in cDPI studies.

In particular, Stankovic-Brandl et al. report an increase in FPF of budesonide at 51% RH compared to 22% RH, using cold-gelled HPMC capsule with a carrier-based formulation [17]. The superior performance of HPMC capsules with carrier-based formulations observed with the phenytoin carrier-based formulation aligns with previous publications. Wauthoz et al. similarly demonstrated superior performance of HPMC capsules, particularly those manufactured by a cold-gelation process, compared with gelatin capsules using formoterol formulations, with no significant differences between Quali-V^®^-I (Qualicaps) and VC capsules at 50% RH [13]. Likewise, Ding et al. found no statistical differences in emitted fraction or FPF among four cold-gelled HPMC capsule types filled with a budesonide carrier-blend formulation tested at 40% RH [18]. Although the impact of electrostatic charges on powder deagglomeration and consequently on DPI performance is already well documented, limited information is available regarding the impact of capsule charging on powder retention upon cDPI device actuation by inhalation (and capsule spinning inside the chamber). In this study, the quantity of electric charges of capsules equilibrated at 45% RH was measured during a flow in contact with ABS, which is the major constituent of the RS01^®^ device. The measured density of charges were all positives, below 2 nC/g for all capsule types, which is consistent with previously reported results by Pinto et al., although stainless steel and PVC were used in their set-up which does not allow direct result comparison [10]. Furthermore, no clear trends could be drawn from the results with lowest values obtained for both gelatin-based capsules but also VCP-I, with very high variability for the latter. Although VCP and VCP-I are constituted of the same polymer and obtained by the same thermo-gelation manufacturing process, they exhibited a different density of charges on ABS and different variability. Plus, the obtained results could not be correlated with phenytoin retention in these capsules after NGI which may indicate that capsule charging in these conditions had limited impact on cDPI performances.

### 4.3. Quality by Design Implications

The batch-to-batch consistency demonstrated by VCP-I capsules (three batches showing no statistically significant differences in key performance parameters) supports their suitability for commercial development under Quality by Design principles. This reproducibility, combined with superior performance, positions VCP-I as the preferred choice for this formulation-device combination. The lower variability also suggests reduced risk of batch failures during commercial manufacturing.

### 4.4. Study Limitations

This study employed a single ternary lactose–phenytoin formulation. Different APIs with varying physicochemical properties (particle size, surface energy, and hygroscopicity) may interact differently with capsule surfaces, potentially altering the performance hierarchy observed. Similarly, results are specific to the RS01 low-resistance device. Different piercing mechanisms, chamber geometries, and airflow patterns in other devices may yield different relative capsule performance rankings.

While 45% RH represents typical storage conditions, real-world usage encompasses a broader range of humidity and temperature conditions that may affect relative capsule performance, particularly given the different moisture sensitivities of gelatin and HPMC materials. The present study was conducted under controlled laboratory which, while representative of standard pharmaceutical manufacturing and testing environments, may not fully capture the complexity of real-world environmental exposures. Clinical use scenarios can involve significantly more challenging conditions, including tropical climates with sustained high humidity (>70% RH), arid environments with very low humidity (<20% RH), and substantial diurnal temperature fluctuations (>15 °C). Such extreme or variable conditions can differentially affect gelatin and HPMC capsule performance through altered moisture content, mechanical properties, and electrostatic behavior.

The current study primarily relied on morphological observations and basic surface measurements. Advanced characterization techniques such as X-ray photoelectron spectroscopy, Dynamic Vapor Sorption analysis, atomic force microscopy, and inverse gas chromatography could provide deeper insights into surface chemistry and energetics relevant to powder–capsule interactions.

### 4.5. Recommendations for Formulation Development

Although the characterization tests performed in this study do not enable a clear understanding of the parameters impacting cDPI capsule performances, the NGI results show that the capsule can have a role on DPI drug product performance. With RS01 device, the designed phenytoin formulation should be filled in a VCP-I capsule to maximize the dose delivered to the lungs. In addition, repeated NGI tests with three different batches of VCP-I confirm the robustness of this capsule, with no statistical differences observed among API retention, EF, FPF, and FPD/nominal.

Based on these findings, formulators developing carrier-based DPI products should consider the following:Capsule Selection Criteria: Prioritize HPMC capsules manufactured via optimized thermo-gelation processes for consistent performance, particularly for moisture-sensitive formulations or products requiring tight dose uniformity specifications.Development Strategy: Include capsule type as a critical material attribute in early formulation screening, as the 10% improvement in deliverable dose observed here could obviate the need for dose increases or formulation modifications.Quality Control: Implement batch-to-batch capsule performance testing as part of incoming material specifications, focusing on reproducibility metrics rather than absolute performance values.Stability Testing Protocol: Design stability protocols that evaluate performance under relevant environmental stress conditions, including temperature–humidity cycling to simulate real-world patient use scenarios. Critical aerodynamic performance attributes (APSD, FPF, FPD, and emitted dose) should be monitored alongside traditional chemical stability endpoints to ensure maintained therapeutic performance under accelerated and long-term storage conditions, with particular focus on moisture-induced changes in inter-particulate forces and powder flowability.

## 5. Conclusions

This study demonstrates that thoughtful capsule selection can significantly improve DPI performance without formulation changes. The consistent performance of VCP-I compared with VCP supports the importance of manufacturing process selection in capsule development and indicates that thermo-gelled capsule shells can perform at least as efficiently as those produced by cold-gelation. These findings provide a scientific basis for capsule selection decisions and highlight the need for continued research into the fundamental mechanisms governing powder-capsule interactions in DPI systems.

## Figures and Tables

**Figure 1 pharmaceutics-17-01621-f001:**
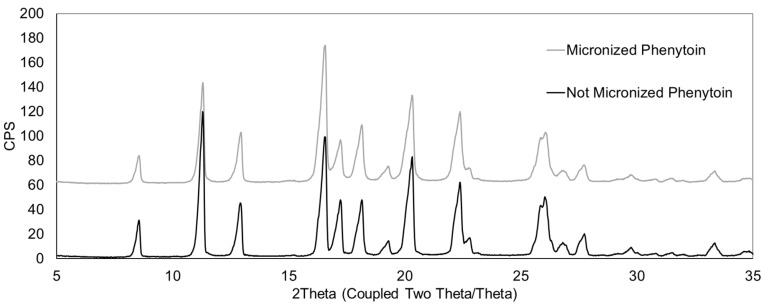
Comparison of XRPD patterns of phenytoin before and after micronization.

**Figure 2 pharmaceutics-17-01621-f002:**
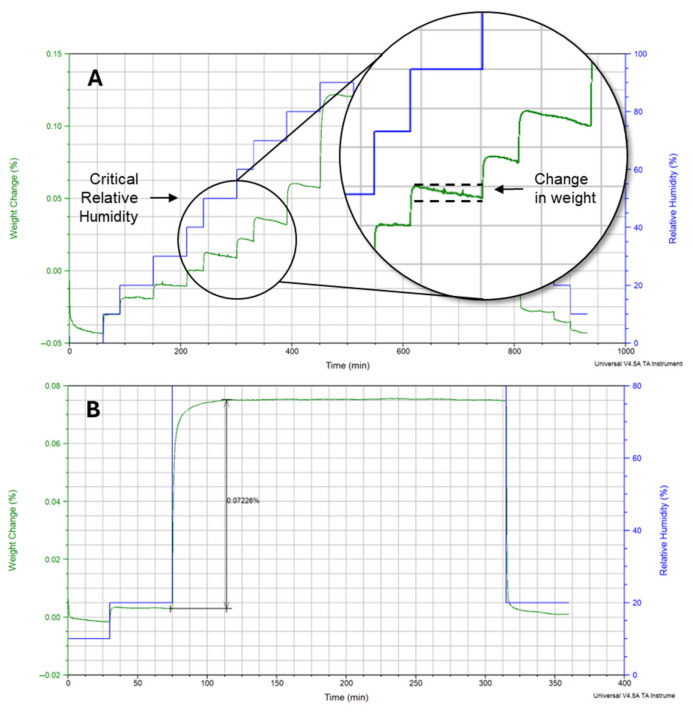
Moisture sorption profiles of phenytoin at increasing relative humidities. (**A**) Weight gain (green) and relative humidity (blue) versus time at 25 °C, with magnified insets highlighting the critical relative humidity region (50% RH) where moisture decrease is observed due to crystallization. (**B**) Sorption–desorption isotherm for micronized phenytoin at 80% RH and 25 °C, showing stable moisture retention, indicating amorphous content is not detected.

**Figure 3 pharmaceutics-17-01621-f003:**
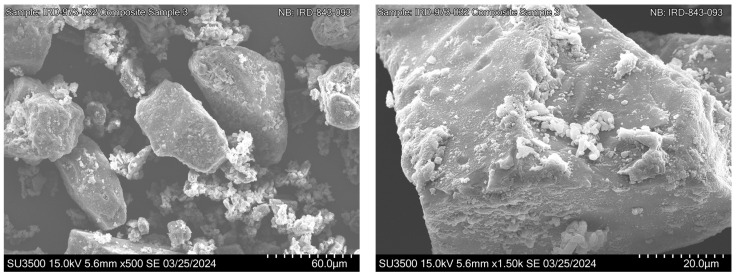
Representative scanning electron micrographs of the phenytoin ternary formulation.

**Figure 4 pharmaceutics-17-01621-f004:**
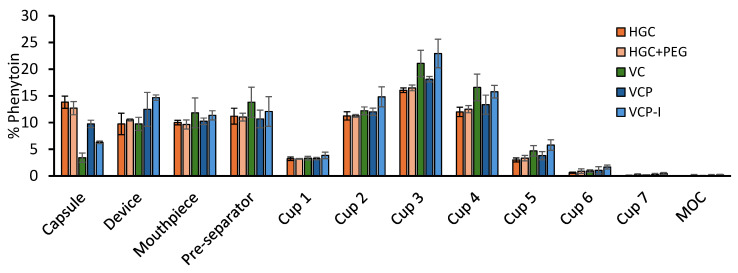
NGI distribution profile of phenytoin formulation filled in various Zephyr^®^ capsules using RS01 device. Results are represented as a percentage of the nominal dose and expressed as a mean ± SD (*n* = 3).

**Figure 5 pharmaceutics-17-01621-f005:**
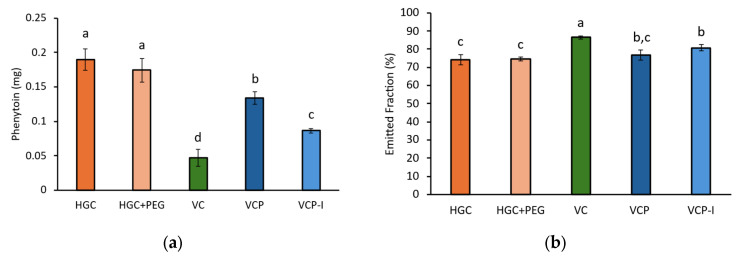
(**a**) Phenytoin quantified in capsule during NGI; (**b**) phenytoin emitted fraction, *n* = 3 per capsule type. Groups represented by the same letter are not significantly different (*p* > 0.05).

**Figure 6 pharmaceutics-17-01621-f006:**
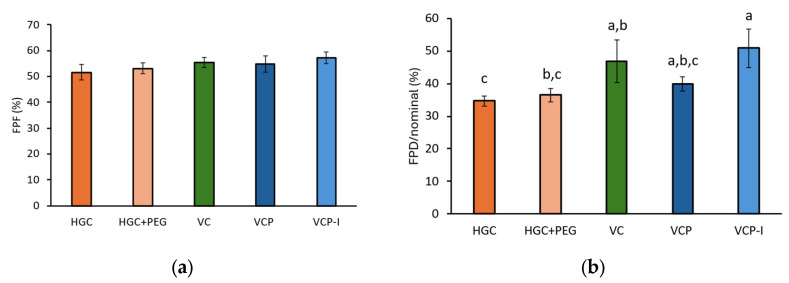
(**a**) Phenytoin FPF, depending on capsule type; (**b**) phenytoin FPD divided by nominal. Groups represented by the same letter are not significantly different (*p* > 0.05).

**Figure 7 pharmaceutics-17-01621-f007:**
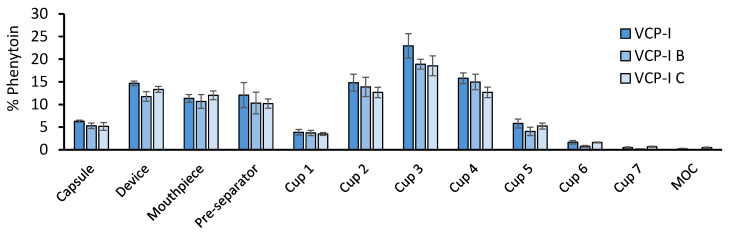
NGI distribution profile of model phenytoin formulation, tested with RS01 device, using three batches of VCP-I capsules, *n* = 3.

**Figure 8 pharmaceutics-17-01621-f008:**
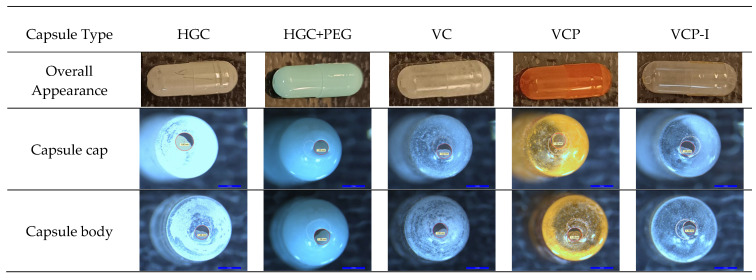
Macroscope observations of size #3 Zephyr^®^ capsules, equilibrated at 45% RH, and emptied from phenytoin formulation using RS01 with DUSA.

**Figure 9 pharmaceutics-17-01621-f009:**
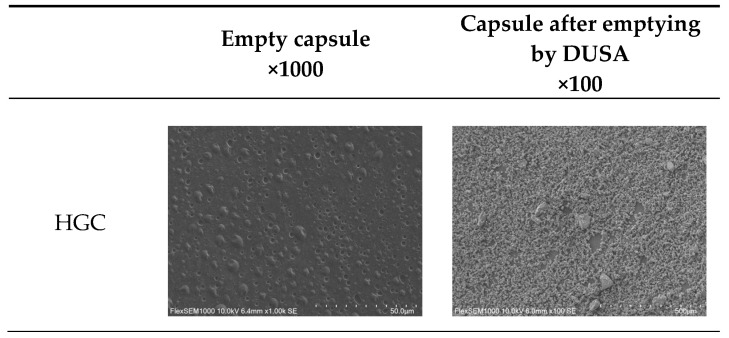
SEM observations of empty size #3 Zephyr^®^ capsules (×1000) and capsules from the same batches emptied from phenytoin formulation using RS01 with DUSA (×100).

**Figure 10 pharmaceutics-17-01621-f010:**
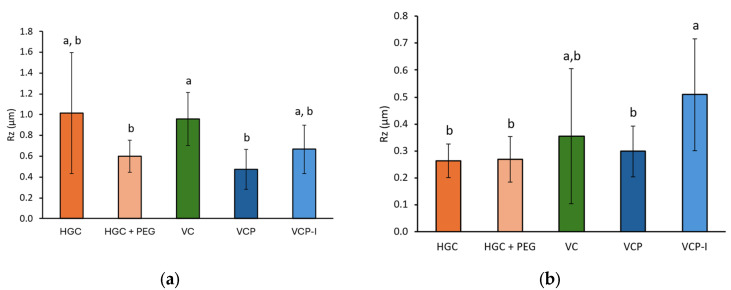
Rugosity measurements performed on the surface of capsule shells in (**a**) stripping direction, *n* = 3 capsules, three measurements per capsule, results expressed as mean ± std dev on the nine values, and (**b**) axial direction, *n* = 3 capsules, six measurements per capsules, results expressed as mean ± std dev on the 18 values. Groups represented by the same letter are not significantly different (*p* > 0.05).

**Figure 11 pharmaceutics-17-01621-f011:**
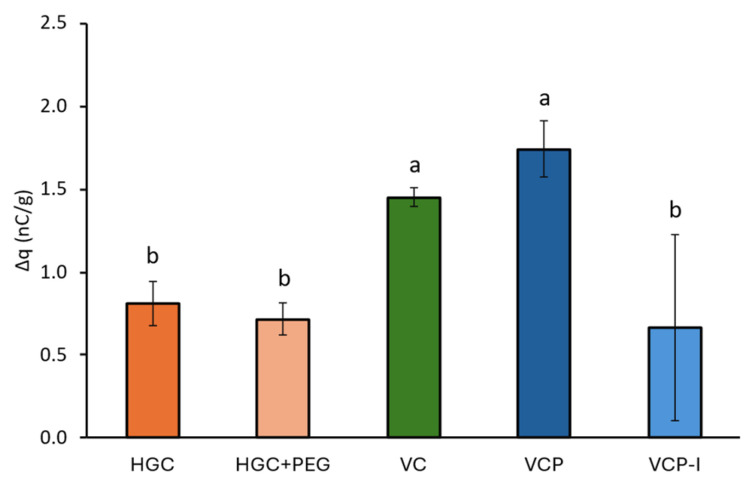
Capsule relative density of charge after contact with ABS, *n* = 3. Groups represented by the same letter are not significantly different (*p* > 0.05).

**Table 1 pharmaceutics-17-01621-t001:** Zephyr^®^ capsules used in the study.

Complete Name	Abbreviation	Polymer	Manufacturing Process	Batch Number	Lub Content (ppm)	Color
Hard Gelatin Capsule	HGC	Gelatin	Cold gelation	3653389	228	Transparent
Hard Gelatin Capsule + PEG	HGC + PEG	Gelatin	Cold gelation	3661448	178	Light green
Vcaps^®^	VC	HPMC	Cold gelation	5424263	209	Transparent
Vcaps^®^ Plus	VCP	HPMC	Thermo-gelation	5425031	174	Orange
Vcaps^®^ Plus Inhance™	VCP-I	HPMC	Thermo-gelation	5417340	46	Transparent
Vcaps^®^ Plus Inhance™ B	VCP-I B	HPMC	Thermo-gelation	5366224	39	Transparent
Vcaps^®^ Plus Inhance™ C	VCP-I C	HPMC	Thermo-gelation	5419911	55	Transparent

**Table 2 pharmaceutics-17-01621-t002:** Aerodynamic properties of phenytoin formulation tested by NGI with three batches of VCP-I and RS01 device, *n* = 3.

APSD Parameters	VCP-I	VCP-I B	VCP-I C
Phenytoin retained in capsule (mg)	0.087 ± 0.003	0.073 ± 0.008	0.071 ± 0.012
EF (%)	81 ± 1.8	82 ± 2.6	81 ± 2
FPF (%)	57 ± 2.2	55 ± 2	55 ± 2
FPD/nominal (%)	51 ± 6	43 ± 4	42 ± 4

**Table 3 pharmaceutics-17-01621-t003:** Capsule water activity after being equilibrated for 10 days at 45% RH, *n* = 3.

Capsule Type	Aw, 21 °C
HGC	0.43 ± 0.00
HGC + PEG	0.43 ± 0.01
VC	0.43 ± 0.01
VCP	0.43 ± 0.01
VCP-I	0.43 ± 0.01

## Data Availability

Data are contained within the article.

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
