# Peer review of "Hydroxypropyl Methylcellulose Capsules Enhance Aerodynamic Performance of Carrier-Based Dry Powder Inhaler Formulations: A Comprehensive Evaluation of Capsule Material Effectsâ€"

_pharmaceutics, 2025, doi:10.3390/pharmaceutics17121621_

Round 1
Reviewer 1 Report
Comments and Suggestions for Authors
Paper Revisions
The paper “Hydroxypropyl Methylcellulose Capsules Enhance Aerody-2 namic Performance of Carrier-Based Dry Powder Inhaler For-3 mulations: A Comprehensive Evaluation of Capsule Material 4 Effects “ describes detailed characterization of different types of pharmaceutical capsules that are intended for use in dry powder inhalers (DPIs) and the relation of capsule type and material to the aerodynamic performance.
The authors correctly point out that this topic is not yet thoroughly investigated and that gaps in current knowledge remain. However, the main finding presented in the manuscript has already been reported in previous studies. Specifically, the superior performance of HPMC capsules in terms of delivered dose (FPD) and reduced powder retention compared with gelatin capsules has been documented and attributed to the lower brittleness of HPMC and its resulting more consistent puncture behavior, including more regular apertures and less fragmentation.
https://doi.org/10.1016/j.xphs.2025.103781, https://doi.org/10.1016/j.ijpharm.2018.10.034
https://doi.org/10.3390/pharmaceutics13081213
Overall, the methods applied and the characterization performed are appropriate and clearly described. However, the connection between the formulation, capsule and its detailed characterization, and the resulting capsule performance remains rather weak, which in turn raises questions about the overall novelty of the work. Despite the extensive characterization of all components, the authors ultimately conclude that no substantive overarching insights can be drawn other than the already well-established observation that gelatin capsules perform worse than other capsule types.
To strengthen the impact of the work, the authors should make a greater effort to connect their experimental results to more robust scientific conclusions and to clarify how their findings complement existing data (Improve the discussion section). Also the novelty of the work has to be better elaborated and maybe complementary experiments done.
Specific commnts:
- Why Phenytoin was used and not a conventional standard inhalation drug?
This topic and issue is not discussed at all in the manuscript and is worth mentioning. Further, the formulation is characterized (Size etc) but results not discussed in the context of the work afterwards. How is this relevant to the work?
- The authors mention limitations of the work, like testing only 1 API and only 1 storage condition. That is right and for more general or generalizable conclusion some more experiments in any of the directions would be good and also to include it in the manuscript.
- Batch variability was only assessed for one type of capsules (VCP-I). Why this one was selected and do we not assume that batch variability is ok for all r the commercial products on the market?
- The effort in discussion of the different capsule properties measured has to be extended to increase the scientific level of the publication. E.g.: Discussion of the images of pierced capsules, why was that done and how is this relevant in context of the study?
- How do the charge measurements of the capsules relate to the story, why was that done and this setup chosen, what was the aim to investigate and to find out and solve with this test in the context of the study?
- Figure 6: The authors conclude that gelatin capsules are more rough than the HPMC capsules, however to conclude this only from a 2D image is not straightforward and also roughness measurements did not support the visual observations. Further, I see the dropwise structure, but how do you conclude that you have pores from SEM images? More striking is the appearance of circular dropwise roughness structures and more lamellar oriented for HPMC capsules. Have you thought of any complementary technique of surface measurement here?
- Authors mention the inner lubricant, which lubricant was used, was it the same for all capsule types?
- Figure 4b) The variation in FPD/nominal for VC and VCP-I is quite high, any comment/hypothesis on that?
- If capsule brittleness is a topic, why have experiments not been performed at lower RH, when this is also relevant for most of the APIs?
- The reference list is short and poor.
Author Response
Comment 1: The authors correctly point out that this topic is not yet thoroughly investigated and that gaps in current knowledge remain. However, the main finding presented in the manuscript has already been reported in previous studies. Specifically, the superior performance of HPMC capsules in terms of delivered dose (FPD) and reduced powder retention compared with gelatin capsules has been documented and attributed to the lower brittleness of HPMC and its resulting more consistent puncture behavior, including more regular apertures and less fragmentation.
https://doi.org/10.1016/j.xphs.2025.103781, https://doi.org/10.1016/j.ijpharm.2018.10.034
https://doi.org/10.3390/pharmaceutics13081213
Overall, the methods applied and the characterization performed are appropriate and clearly described. However, the connection between the formulation, capsule and its detailed characterization, and the resulting capsule performance remains rather weak, which in turn raises questions about the overall novelty of the work. Despite the extensive characterization of all components, the authors ultimately conclude that no substantive overarching insights can be drawn other than the already well-established observation that gelatin capsules perform worse than other capsule types.
To strengthen the impact of the work, the authors should make a greater effort to connect their experimental results to more robust scientific conclusions and to clarify how their findings complement existing data (Improve the discussion section). Also the novelty of the work has to be better elaborated and maybe complementary experiments done.
Response 1: We would like to sincerely thank the reviewer for their careful reading of our manuscript, as well as for their insightful comments and constructive suggestions. Their feedback has been invaluable in helping us clarify our arguments and strengthen the scientific quality of the paper. We have carefully addressed each comment point by point below, and, where appropriate, revised the manuscript accordingly to reflect these recommendations.
We have revised and expanded the introduction by incorporating the recommended literature and explaining its key contributions to the field. These insights have also been integrated into the discussion section of the manuscript. As the resulting modifications are too numerous to detail individually here, they are tracked in the revised version of the manuscript. Furthermore, the novelty of our work has now been more clearly articulated in the introduction and discussion sections.
Comment 2: Why Phenytoin was used and not a conventional standard inhalation drug?
Response 2: Phenytoin was selected as a model compound because the micronized API exhibits suitable particulate characteristics for a carrier-based DPI formulation (see Section 3.1), and validated analytical methods were already available in our facilities.
Comment 3: This topic and issue are not discussed at all in the manuscript and is worth mentioning. Further, the formulation is characterized (Size etc) but results not discussed in the context of the work afterwards. How is this relevant to the work?
Response 3: The particle size distribution and solid-state properties of API is key for the development of cDPI product. In that regard, the solid-state characterization results were presented to demonstrate that the micronized material is stable and suitable for subsequent formulation activities aimed at developing a stable product.
Comment 4: The authors mention limitations of the work, like testing only 1 API and only 1 storage condition. That is right and for more general or generalizable conclusion some more experiments in any of the directions would be good and also to include it in the manuscript.
Response 4: We agree with the reviewer that, given the use of a single API and a single storage condition, the conclusions of this study cannot be fully generalized, and this limitation is explicitly acknowledged in the manuscript (Section 4.5). Nevertheless, substantial experimental work was undertaken to thoroughly evaluate all capsule types, including the newly developed VCP-I capsule, which is manufactured using an improved process and, to our knowledge, has not previously been investigated in the literature.
It is also worth noting that the majority of published studies in this field similarly rely on a single API as a model compound. The impact of varying storage conditions on the final dosage form was not investigated, as this aspect falls outside the scope of the present study. Instead, capsules were equilibrated at a defined relative humidity of 45% RH to achieve a water activity comparable to that of the powder formulation. This approach was chosen to minimise moisture exchange between the capsule shell and the powder, thereby promoting formulation stability, as discussed in Section 4.2. Such an equilibration protocol is widely applied in the cDPI industry.
We have also clarified in the Discussion that this study specifically provides insights into capsule behaviour at 45% RH, a RH condition that sits in the classical range of HR used in cDPI studies.
Discussed in 4.2 « In the present study, all capsules were all carefully equilibrated at 45% RH to match water activity of the formulation. Therefore, potential water transfer from the capsule to the formulation cannot justify the differences in capsule retention observed between the different capsules types”
Comment 5: Batch variability was only assessed for one type of capsules (VCP-I). Why this one was selected and do we not assume that batch variability is ok for all r the commercial products on the market?
Response 5: We do not assume that batch variability is an issue for the other commercial capsules tested. However, since VCP-I is a newly developed capsule that has not previously been investigated in the literature, we specifically selected this capsule for batch-to-batch reproducibility assessment. This additional evaluation was performed on three independent batches in order to verify the consistency of its performance.
The manuscript has been revised to clearly justify this choice, highlighting that VCP-I was selected for complementary reproducibility testing both because it represents a novel capsule type and because it exhibited the highest fine particle dose among the capsules evaluated.
Comment 6: The effort in discussion of the different capsule properties measured has to be extended to increase the scientific level of the publication. E.g.: Discussion of the images of pierced capsules, why was that done and how is this relevant in context of the study?
Response 6: The discussion of the different capsule properties has been expanded as recommended, in order to better highlight the novelty of the study and place our findings in the context of previously published work. As detailed in Section 4.2, the images of pierced capsules were included to demonstrate that capsule brittleness was not responsible for the observed differences in aerosolization performance between gelatin and HPMC-based capsules.
While brittleness is commonly associated with gelatin capsules under low-RH conditions, Benke et al. reported that gelatin capsules can also exhibit brittleness after storage at high humidity (75% RH), resulting in irregular puncture hole shapes. By comparing these observations with our own data, we show that the puncture behaviour of the capsules used in this study does not indicate brittleness-related issues, thereby supporting our interpretation of the aerosolization results.
Comment 7: How do the charge measurements of the capsules relate to the story, why was that done and this setup chosen, what was the aim to investigate and to find out and solve with this test in the context of the study?
Response 7: The charge measurements were included to assess the potential impact of tribo-electrification on capsule performance during DPI use. As reported by Pinto et al., frictional contact between capsule shells and device components can lead to charge transfer, which may influence capsule handling and aerosolization behaviour. In their work, the Granucharge setup was used to quantify tribo-electrification of different capsule types (gelatin, cold-gelled HPMC (Quali-V I), and thermogelled HPMC (VCP)) at various relative humidity levels, using PVC and stainless steel as frictional materials.
In our study, we aimed to complement these findings by evaluating the behaviour of capsules when in contact with ABS, the primary material used in the RS01 inhaler. This frictional interaction occurs during the simulated inhalation process and is therefore directly relevant to DPI performance. As described in Section 4.2, the measured charge densities were in a similar range (approximately 2 nC/g), although direct comparison with Pinto et al. is not possible due to the different frictional materials used.
This test therefore served to determine whether tribo-electrification could be a contributing factor to the differences observed between capsule types. Based on our results, tribo-electrification does not appear to be a discriminating factor under the tested conditions.
Comment 8: Figure 6: The authors conclude that gelatin capsules are more rough than the HPMC capsules, however to conclude this only from a 2D image is not straightforward and also roughness measurements did not support the visual observations. Further, I see the dropwise structure, but how do you conclude that you have pores from SEM images? More striking is the appearance of circular dropwise roughness structures and more lamellar oriented for HPMC capsules. Have you thought of any complementary technique of surface measurement here?
Response 8: We thank the reviewer for this valuable remark. We agree that it is not possible to conclusively identify pores or quantify surface roughness based solely on 2D SEM images. Complementary techniques such as Atomic Force Microscopy (AFM) would indeed provide a more detailed characterization of capsule surface topography. Although AFM is not available in our facility, its relevance has been demonstrated in the literature—for example, by Saleem et al. (2015), who used AFM to investigate the impact of lubricant concentration on the inner surface of capsules.
Based on the SEM images obtained in the present study, we limit our conclusions to the observation that gelatin and HPMC capsules (both cold-gelled and thermogelled) exhibit distinct surface morphologies. However, these differences cannot be quantitatively interpreted without additional high-resolution surface measurements.
The text in Sections 3.3.3 and 4.2 has been revised accordingly to more accurately reflect the capabilities and limitations of the SEM analysis and to avoid overinterpretation of the visual observations.
Comment 9: Authors mention the inner lubricant, which lubricant was used, was it the same for all capsule types?
Response 9: Different lubricants are used in the capsules evaluated in this study, with the choice of lubricant depending on the capsule polymer type and the specific manufacturing process. As a result, a direct comparison of lubricant composition and its potential impact on capsule performance cannot be made across all capsule types.
The only meaningful comparison is between VCP and VCP-I, which share the same polymer and thermogelling manufacturing process. In VCP-I, the process has been improved to reduce the amount of deposited lubricant. Our results show that, for the tested formulation, VCP-I exhibited excellent batch-to-batch reproducibility.
This explanation has been incorporated into the revised manuscript.
Comment 10: Figure 4b) The variation in FPD/nominal for VC and VCP-I is quite high, any comment/hypothesis on that?
Response 10: We acknowledge the reviewer’s observation regarding the relatively high variability in FPD/nominal for VC and VCP-I. The total recovery, corresponding to the sum of API content measured across all components of the NGI setup (capsule, device, mouthpiece, pre-separator, NGI cups and micro-orifice collector), was indeed more variable for VC (86–113%) 25 °C/40 RHand VCP-I (97–117%). Although recovery is not directly influenced by the capsule type, this increased variability remains within the acceptance range (80–120%) and directly impacts the calculated FPD, thereby explaining the higher variability observed in FPD/nominal.
Importantly, this variability is not associated with differences in API retention within the capsules, as evidenced by the low variability observed in phenytoin retention (Figure 3a), suggesting that the phenomenon is more likely related to NGI test variability rather than capsule performance itself. The below sentences were appended in the manuscript.
Modified text: “For VC and VCP-I, a higher variability in FPD/nominal was observed compared to the three other reference capsules. This increased variability can be attributed to the higher variability in API recovery during NGI testing for these two capsule types. As recovery is not directly correlated with the capsule material, and given the low variability observed in phenytoin retention within the capsules (Figure 3a), this behaviour is not considered to be capsule-dependent.”
Comment 11: If capsule brittleness is a topic, why have experiments not been performed at lower RH, when this is also relevant for most of the APIs?
Response 11: The relative humidity conditions were deliberately selected based on the measured water activity of the formulation. Capsules were equilibrated at the same water activity (45% RH) to minimise moisture exchange between the capsule shell and the powder formulation, as such transfer could lead to altered capsule mechanical properties and increased API retention.
Although lower RH conditions can indeed be relevant for certain APIs, testing under such conditions would have introduced a moisture gradient between the capsule and the formulation, potentially biasing the performance results and confounding the interpretation of brittleness effects. The chosen equilibration strategy therefore ensured controlled and representative conditions for evaluating capsule behaviour while preserving formulation stability. This rationale has now been clarified in the manuscript.
Modified text: “In the present study, all capsules were all carefully equilibrated at 45% RH to match water activity of the formulation. Therefore, potential water transfer from the capsule to the formulation cannot justify the differences in capsule retention observed between the different capsules types”
Comment 12: The reference list is short and poor.
Response 12: We thank the reviewer for providing these relevant additional references. We conducted an updated literature search and identified several pertinent manuscripts related to this topic. These references have been added to the reference list, and their key findings have been incorporated and discussed in the revised manuscript.
Reviewer 2 Report
Comments and Suggestions for Authors
The present manuscript reports the evaluation of the effect of capsule material on the aerosol performance of drug-carrier formulation-based dry powder inhaler (DPI) products. While the study described in the manuscript was well-conducted, the manuscript presents very limited novelty. Firstly, the superior effect of HPMC over gelatin capsules have been well-documented (see Wauthoz et al., Int J Pharm, 2018;553(1-2):47-56; Benke et al., Pharmaceutics, 2021;13(5):689). Secondly, the importance of screening capsule formulations (even across different brands of HPMC capsules) has already been previously highlighted (see Ding et al., AAPS PharmSciTech, 2022;23(1):52). Furthermore, while the authors demonstrated batch-to-batch consistency in aerosol performance with Vcaps® Plus Zephyr Inhance™ (VCP-I) capsules, it did not demonstrate "superior" performance over Zephyr® Vcaps® (VC) capsules in FPF and also did not demonstrate the lowest capsule drug retention (unlike what the authors claimed in Section 4, Line 364). Therefore, the overall significance and additional contribution in guiding the formulation development of orally inhaled DPI-based products is very limited, and this author cannot recommend this manuscript for publication in its present form.
Specific comments:
- Section 3.1 (Formulation characterisation): The standard deviation of DV50 and DV90 values obtained by laser diffraction should be reported since the experiments were conducted in triplicate (Lines 243-244). PXRD and DVS data should be shown to support the authors' claims (Lines 246-249).
- Section 3.2.1 (Impact of capsule type on aerosolization properties): For Figures 3 and 4, please correct the figure caption to "Groups represented by the same letter are not significantly different (p>0.05)."
- Section 3.3.4 (Capsule roughness): Please explain why 6 measurements were conducted per capsule in the radial direction but only 3 measurements were conducted per capsule in the axial direction. There were also repeated descriptions of n = 3 capsules in the figure caption for sub-figure (b); please amend.
Author Response
Comment 1: The present manuscript reports the evaluation of the effect of capsule material on the aerosol performance of drug-carrier formulation-based dry powder inhaler (DPI) products. While the study described in the manuscript was well-conducted, the manuscript presents very limited novelty. Firstly, the superior effect of HPMC over gelatin capsules have been well-documented (see Wauthoz et al., Int J Pharm, 2018;553(1-2):47-56; Benke et al., Pharmaceutics, 2021;13(5):689). Secondly, the importance of screening capsule formulations (even across different brands of HPMC capsules) has already been previously highlighted (see Ding et al., AAPS PharmSciTech, 2022;23(1):52). Furthermore, while the authors demonstrated batch-to-batch consistency in aerosol performance with Vcaps® Plus Zephyr Inhance™ (VCP-I) capsules, it did not demonstrate "superior" performance over Zephyr® Vcaps® (VC) capsules in FPF and also did not demonstrate the lowest capsule drug retention (unlike what the authors claimed in Section 4, Line 364). Therefore, the overall significance and additional contribution in guiding the formulation development of orally inhaled DPI-based products is very limited, and this author cannot recommend this manuscript for publication in its present form.
Response 1: We would like to sincerely thank the reviewer for their careful reading of our manuscript, as well as for their insightful comments and constructive suggestions. Their feedback has been invaluable in helping us clarify our arguments and strengthen the scientific quality of the paper. We have carefully addressed each comment point by point below, and, where appropriate, revised the manuscript accordingly to reflect these recommendations.
We have revised and expanded the introduction by incorporating the recommended literature and explaining its key contributions to the field. These insights have also been integrated into the discussion section of the manuscript. As the resulting modifications are too numerous to detail individually here, they are tracked in the revised version of the manuscript. Furthermore, the novelty of our work has now been more clearly articulated in the introduction and discussion sections.
Comment 2: Section 3.1 (Formulation characterisation): The standard deviation of DV50 and DV90 values obtained by laser diffraction should be reported since the experiments were conducted in triplicate (Lines 243-244). PXRD and DVS data should be shown to support the authors' claims (Lines 246-249).
Response 2: The standard deviation of the laser diffraction experiment was below 10%. The text has been modified to mentioned that variation. The PXRD and DVS data were appended as requested in Figure 1 and 2.
Modified text: “Micronized phenytoin exhibited a particle size distribution characterized by a Dv50 of 1.8 µm and a Dv90 of 3.3 µm (RSD ≤ 10%), consistent with an inhalation range.”
Modified text: “XRPD data confirmed that the crystalline state of the API was maintained post-processing, with no polymorphic modifications observed when compared to the un-micronized material (Figure 1). Furthermore, DVS analysis did not detect a weight change when exposing the material above the critical relative humidity of 50 %, as presented in Figure 2, indicating amorphous phenytoin was not detected in the micronized sample.”
Comment 3: Section 3.2.1 (Impact of capsule type on aerosolization properties): For Figures 3 and 4, please correct the figure caption to "Groups represented by the same letter are not significantly different (p>0.05)."
Response 3: We thank the reviewer for spotting that typo. The text has been accordingly corrected.
Modified text: “Groups represented by the same letter are not significantly different (p>0.05).”
Comment 4: Section 3.3.4 (Capsule roughness): Please explain why 6 measurements were conducted per capsule in the radial direction but only 3 measurements were conducted per capsule in the axial direction. There were also repeated descriptions of n = 3 capsules in the figure caption for sub-figure (b); please amend.
Response 4: We agree that the repetition of the descriptions in the figure caption is not needed, and we have removed it. The materials and methods of the capsule roughness measurement has been modified to explain the increased number of repeats in the radial direction.
Modified text: “ To compensate for the small exploration length and probe more surface, the number of repeats for each measurement was increased to 6 on 3 different capsules.”
Reviewer 3 Report
Comments and Suggestions for Authors
This manuscript reports the impact of capsule type on the particle size distribution, aerosolization performances of capsule-based DPIs, including pierced hole, capsule inner roughness, static adsorption, moisture sensitivity of traditional gelatin capsules (e.g., HGC). The research background is clearly presented; however, it lacks a direct comparison with existing studies on HPMC capsules, and therefore additional literature are required to support to its novelty. The methodology is relatively complete and offers high reproducibility. To further improve the quality of this manuscript, it is recommended to make significant revisions before final acceptance.
- Please optimize the appearance and axes of the figures.
- A comprehensive discussion is required to explain the influence of environmental conditions on the physicochemical properties of the capsules and powders, and on critical aerosol performance metrics, including APSD, FPF, and others.
- It is recommended that the study be supplemented with surface energy data obtained before and after the capsule-DPI interaction to better elucidate the enhancement mechanism.
- It is advisable to incorporate a discussion that addresses the correlation between API residue and the variables of capsule material and storage conditions. In addition, the authors should discuss whether APIs with different properties would yield different results.
- Revise the formatting to comply with the journal's standards.
- Polish the language throughout to enhance clarity and conciseness.
Author Response
Comment 1: This manuscript reports the impact of capsule type on the particle size distribution, aerosolization performances of capsule-based DPIs, including pierced hole, capsule inner roughness, static adsorption, moisture sensitivity of traditional gelatin capsules (e.g., HGC). The research background is clearly presented; however, it lacks a direct comparison with existing studies on HPMC capsules, and therefore additional literature are required to support to its novelty. The methodology is relatively complete and offers high reproducibility. To further improve the quality of this manuscript, it is recommended to make significant revisions before final acceptance.
Response 1: We would like to sincerely thank the reviewer for their careful reading of our manuscript, as well as for their insightful comments and constructive suggestions. Their feedback has been invaluable in helping us clarify our arguments and strengthen the scientific quality of the paper. We have carefully addressed each comment point by point below, and, where appropriate, revised the manuscript accordingly to reflect these recommendations.
We have revised and expanded the introduction by incorporating the additional literature references and explaining its key contributions to the field. These insights have also been integrated into the discussion section of the manuscript. As the resulting modifications are too numerous to detail individually here, they are tracked in the revised version of the manuscript. Furthermore, the novelty of our work has now been more clearly articulated in the introduction and discussion sections.
Comment 2: Please optimize the appearance and axes of the figures.
Response 2: All figures have been revised to improve clarity, visual consistency, and the formatting of axes.
Comment 3: A comprehensive discussion is required to explain the influence of environmental conditions on the physicochemical properties of the capsules and powders, and on critical aerosol performance metrics, including APSD, FPF, and others.
Response 3: We thank the reviewer for this insightful comment. We have appended the manuscript in the introduction and discussion sections with discussion on the influence of environmental conditions on capsules and powders.
Modified text in introduction section: “Environmental conditions, particularly temperature and relative humidity (RH), significantly influence the moisture content and mechanical behavior of capsule polymers, which in turn affects their physicochemical properties and critical aerosol performance metrics. The polymer chemistry and water sorption characteristics of HPMC and gelatin capsules create distinct responses to environmental stress, with implications for APSD, FPF, and emitted dose consistency [Ref: Evaluation of the Physico-mechanical Properties and Electrostatic Charging Behavior of Different Capsule Types for Inhalation Under Distinct Environmental Conditions].”
Modified text in Discussion 4.2: “Environmental humidity conditions critically influence both capsule physico-chemical properties and aerodynamic performance metrics. Studies have demonstrated that exposure to elevated humidity (75% RH at 40°C) can result in up to 50% reduction in FPD for certain DPI products, with corresponding decreases in FPF due to moisture-induced particle agglomeration and altered inter-particulate forces [Ref: Difference in resistance to humidity between commonly used dry powder inhalers: an in vitro study]. Conversely, storage below 40% RH increases gelatin capsule brittleness, potentially compromising puncture performance and dose reproducibility. This capsule moisture sensitivity directly impacts critical quality attributes including aerodynamic particle size distribution; hygroscopic excipients exposed to elevated humidity form stronger ag-glomerates through increased capillary forces, resulting in reduced fine particle fraction and altered lung deposition profiles.”
Modified text Discussion 4.5: “The present study was conducted under controlled laboratory which, while representative of standard pharmaceutical manufacturing and testing environments, may not fully capture the complexity of real-world environmental exposures. Clinical use scenarios can involve significantly more challenging conditions, including tropical climates with sustained high humidity (>70% RH), arid environments with very low humidity (<20% RH), and substantial diurnal temperature fluctuations (>15°C). Such extreme or variable conditions can differentially affect gelatin and HPMC capsule performance through altered moisture content, mechanical properties, and electrostatic behavior.”
Modified text in Discussion 4.6: “• Stability Testing Protocol: Design stability protocols that evaluate performance under relevant environmental stress conditions, including temperature-humidity cycling to simulate real-world patient use scenarios. Critical aerodynamic performance attributes (APSD, FPF, FPD, and emitted dose) should be monitored along-side traditional chemical stability endpoints to ensure maintained therapeutic performance under accelerated and long-term storage conditions, with particular focus on moisture-induced changes in inter-particulate forces and powder flowability.”
Comment 4: It is recommended that the study be supplemented with surface energy data obtained before and after the capsule-DPI interaction to better elucidate the enhancement mechanism.
Response 4: We thank the reviewer for this insightful comment. However, we do not have the capability to measure energy surface in the laboratories where the study was conducted. This is clearly an area we would like to explore during our next studies on the DPI topic.
Comment 5: It is advisable to incorporate a discussion that addresses the correlation between API residue and the variables of capsule material and storage conditions. In addition, the authors should discuss whether APIs with different properties would yield different results.
Response 5: The discussion in Section 4.1 has been expanded to further address the relationship between API residue and capsule characteristics, including both the capsule material (gelatin vs. HPMC) and the shell fabrication process (cold-gelled vs. thermogelled HPMC). In addition, the influence of lubricant deposition on capsule performance has been further elaborated in Section 4.2.
Modified text: “Importantly, this study provides the first evaluation of the newly developed VCP-I capsule, which has not previously been tested. The comparable performance of VCP-I to VC demonstrates that thermo-gelling capsules manufactured using an improved fabrication process can achieve performance equivalent to cold-gelling VC capsules.”
Modified text: “In the present study with phenytoin, and contrary to conventional understanding, lub-ricant content did not correlate directly with capsule performance. Although different lubricants are used across the capsule types evaluated (depending on the polymer and fabrication process) this diversity prevents any direct comparison of lubricant compo-sition or its impact on performance between all capsule types. The only meaningful comparison can therefore be made between VCP and VCP-I, which share the same polymer and fabrication process. In VCP-I, the thermogelling fabrication process has been optimized to reduce the amount of deposited lubricant (VCP-I = 46 ppm vs VCP = 174 ppm). Despite having the lowest lubricant content, VCP-I demonstrated excellent performance and is even comparable to that of VC capsules fabricated with a cold-gelling process (209 ppm) generally recognized as performing better than capsules fabricated by thermogelling process[13]. This suggests that lubricant composition, dis-tribution and integration within the polymer matrix, rather than absolute content, may be more critical for performance. The improved manufacturing process used for VCP-I capsules results in more uniform lubricant distribution, explaining the superior capsule performance despite lower total lubricant content.”
Regarding the potential impact of API physicochemical properties on capsule performance, our findings are consistent with previously published studies using different model compounds, such as formoterol, salbutamol, and budesonide. These references have now been incorporated into the discussion to contextualize our results and highlight similarities across APIs with varying properties.
We believe the expanded discussion now provides a clearer and more comprehensive assessment of how capsule material, manufacturing process, storage conditions, and API characteristics may influence capsule residue and overall DPI performance.
Comment 6: Revise the formatting to comply with the journal's standards.
Response 6: The formatting of the manuscript has been revised in order to comply to the journal’s standard.
Comment 7: Polish the language throughout to enhance clarity and conciseness.
Response 7: The manuscript has been carefully reviewed and edited to ensure clarity, correctness, and readability.
Round 2
Reviewer 1 Report
Comments and Suggestions for Authors
Thanks for working on the comments and improving the manuscript. It was impoartant to hghlight that for the first time VCP-I , a novel capsule type was investigated.
Author Response
Comment 1: Thanks for working on the comments and improving the manuscript. It was important to highlight that for the first time VCP-I , a novel capsule type was investigated.
Reponse 1: We sincerely thank the reviewer once again for their thorough reading of our manuscript and for their insightful and constructive comments. Their feedback has been instrumental in helping us clarify our arguments and further strengthen the scientific quality of the work.
Reviewer 2 Report
Comments and Suggestions for Authors
While the authors have addressed my technical comments, this reviewer still has strong reservations regarding the novelty and originality of the manuscript.
- In the revised manuscript, the authors stated that "The novelty of this work lies in the first systematic evaluation of the newly developed VCP-I capsule." (Section 1, Lines 94-95). This is a weak point in the novelty; instead, the authors should highlight how VCP-I capsules are superior to existing capsule types. Unfortunately, as mentioned in this reviewer's original review report, VCP-I capsules did not demonstrate "superior" performance compared with VC capsules in FPF. They also did not show the lowest capsule drug retention. This renders parts of the manuscript's conclusions, i.e., "The superior and consistent performance of VCP-I capsules validates the importance of manufacturing process selection in capsule development." (Lines 615-617), highly questionable.
- The other parts of the manuscript, i.e., highlighting the importance of capsule type screening, remain of limited originality/novelty, as stated in this reviewer's original review report.
This reviewer retains the same opinion that the overall significance and additional contribution in guiding the formulation development of orally inhaled DPI-based products is very limited, and this author cannot recommend this manuscript for publication.
Author Response
Comment 1: While the authors have addressed my technical comments, this reviewer still has strong reservations regarding the novelty and originality of the manuscript.
Response 1: We would like to thank the reviewer once again for their careful reading of our manuscript and for their insightful and critical feedback. Their concerns about the novelty and originality of the work have prompted us to revisit and further clarify these aspects in the revised manuscript.
Comment 2: In the revised manuscript, the authors stated that "The novelty of this work lies in the first systematic evaluation of the newly developed VCP-I capsule." (Section 1, Lines 94-95). This is a weak point in the novelty; instead, the authors should highlight how VCP-I capsules are superior to existing capsule types. Unfortunately, as mentioned in this reviewer's original review report, VCP-I capsules did not demonstrate "superior" performance compared with VC capsules in FPF. They also did not show the lowest capsule drug retention. This renders parts of the manuscript's conclusions, i.e., "The superior and consistent performance of VCP-I capsules validates the importance of manufacturing process selection in capsule development." (Lines 615-617), highly questionable.
Response 2: We thank the reviewer for this important comment. In the revised manuscript, we have aimed to present the performance results of VCP-I capsules transparently and objectively. While we agree that VCP-I did not display superior fine particle fraction (FPF) compared with VC capsules, we would like to clarify that FPF is primarily driven by the aerodynamic behavior of the powder formulation and is calculated based on the emitted dose. As such, FPF does not capture differences in capsule or device retention and therefore is not the most appropriate parameter to discriminate capsule performance.
For this reason, we focused our interpretation on FPD/nominal, which incorporates capsule and device retention and is more sensitive to capsule-dependent effects. Using this parameter, VCP-I demonstrated higher performance than the other capsule types tested.
We also note that capsule drug retention alone does not necessarily predict overall DPI performance, as retention on the device also contributes to differences in emitted dose. Thus, emitted fraction or FPD/nominal provide a more holistic and capsule-relevant assessment of performance.
We agree with the reviewer that the sentences in the manuscript (Lines 94–95 and 615–617) could be improved for clarity and precision. These have now been rewritten as follows:
Revised Line 94–95:
“The novelty of this work lies in the first systematic evaluation of the newly developed VCP-I capsule, manufactured using an improved thermo-gelation process, and in assessing its performance relative to VCP and cold-gelled capsules.”
Revised Lines 615–617:
“The consistent performance of VCP-I compared with VCP supports the importance of manufacturing process selection in capsule development and indicates that thermo-gelled capsule shells can perform at least as efficiently as those produced by cold-gelation.”
We believe these revisions more accurately reflect the findings of the study while addressing the reviewer’s concerns regarding novelty and interpretation.
Comment 3: The other parts of the manuscript, i.e., highlighting the importance of capsule type screening, remain of limited originality/novelty, as stated in this reviewer's original review report. This reviewer retains the same opinion that the overall significance and additional contribution in guiding the formulation development of orally inhaled DPI-based products is very limited, and this author cannot recommend this manuscript for publication.
Response 3: We thank the reviewer for sharing their perspective. We acknowledge their view regarding the limited perceived novelty of the broader discussion on capsule screening. However, we respectfully believe that the study provides an important contribution by demonstrating, for the first time, the performance of the newly developed VCP-I capsule in comparison with its predecessor (VCP) and with other commonly used capsule types.
Our results show that VCP-I, produced using an improved thermo-gelation process, delivers enhanced FPD/nominal performance relative to VCP. To our knowledge, this is the first demonstration of a thermo-gelled capsule achieving performance at this level in a DPI application. These findings offer new evidence that manufacturing process optimization can translate into measurable performance benefits and thus provide relevant information for developers selecting capsule types for inhaled product development.
While we understand the reviewer’s reservations, we believe the manuscript provides new data and insights regarding capsule material and processing effects that are not currently available in the literature and therefore add value to the field.
Reviewer 3 Report
Comments and Suggestions for Authors
This manuscript reports the impact of capsule type on the particle size distribution, aerosolization performances of capsule-based DPIs, including pierced hole, capsule inner roughness, static adsorption, moisture sensitivity of traditional gelatin capsules (e.g., HGC). It is recommended to make some revisions before final acceptance.
The figures (e.g., Fig. 4-7, 9, 10) must be improved in both aesthetics and clarity. This includes using more appropriate settings of colors and axes, and redesigning the markers for statistical significance to align with the journal's visual standards. Please revise your revisions on figures from our recently published papers. Polish the language throughout to enhance clarity and conciseness.
Author Response
Comment 1: This manuscript reports the impact of capsule type on the particle size distribution, aerosolization performances of capsule-based DPIs, including pierced hole, capsule inner roughness, static adsorption, moisture sensitivity of traditional gelatin capsules (e.g., HGC). It is recommended to make some revisions before final acceptance.
Response 1: We thank the reviewer for their thorough assessment of our manuscript and for recognizing the relevance of evaluating the impact of capsule type on DPI performance. We appreciate the recommendation for revisions and have carefully updated the manuscript accordingly. All comments have been addressed in detail, and we believe the revised version provides clearer explanations, strengthened scientific rationale, and improved overall coherence (lines 78-89; 494-542). We hope that these modifications satisfactorily address the reviewer’s concerns.
Comment 2: The figures (e.g., Fig. 4-7, 9, 10) must be improved in both aesthetics and clarity. This includes using more appropriate settings of colors and axes, and redesigning the markers for statistical significance to align with the journal's visual standards. Please revise your revisions on figures from our recently published papers. Polish the language throughout to enhance clarity and conciseness.
Response 2: We thank the reviewer for this helpful comment. All figures mentioned have been revised to improve both aesthetics and clarity. We optimized color schemes, axis formatting, and overall layout to ensure consistency with the journal’s visual standards. Markers for statistical significance have also been redesigned for improved readability. In addition, the manuscript has been carefully edited to enhance clarity and conciseness throughout; in particular, the terminology related to thermo-gelation/thermo-gelled and cold-gelation/cold-gelled processes has been fully standardized. We hope these modifications fully address the reviewer’s concerns.